# Improving the accuracy of automated labeling of specimen images datasets via a confidence-based process

Quentin Bateux[1], Jonathan Koss[2], Patrick W. Sweeney[3], Erika Edwards[4], Nelson Rios[5], Aaron M. Dollar[6]*

1 Institute for Biospheric Studies, Yale University, New Haven, Connecticut, United States of America, 2 Department of Electrical Engineering, Yale University, New Haven, Connecticut, United States of America, 3 Division of Botany at the Yale Peabody Museum, Yale University, New Haven, Connecticut, United States of America, 4 Department of Ecology and Evolutionary Biology, Yale University, New Haven, Connecticut, United States of America, 5 Peabody Museum and Yale School of Forestry and Environmental Studies, Yale University, New Haven, Connecticut, United States of America, 6 Department of Mechanical Engineering and Materials Science, Yale University, New Haven, Connecticut, United States of America

* aaron.dollar@yale.edu

## Abstract

The digitization of natural history collections over the past three decades has unlocked a treasure trove of specimen imagery and metadata. There is great interest in making this data more useful by further labeling it with additional trait data, and modern "deep learning" machine learning techniques utilizing convolutional neural nets (CNNs) and similar networks show particular promise to reduce the amount of required manual labeling by human experts, making the process much faster and less expensive. However, in most cases, the accuracy of these approaches is too low for reliable utilization of the automatic labeling, typically in the range of 80-85% accuracy. In this paper, we present and validate an approach that can greatly improve this accuracy, essentially by examining the "confidence" that the network has in the generated label as well as utilizing a user-defined threshold to reject labels that fall below a chosen level. We demonstrate that a naive model that produced 86% initial accuracy can achieve improved performance - over 95% accuracy (rejecting about 40% of the labels) or over 99% accuracy (rejecting about 65%) by selecting higher confidence thresholds. This gives flexibility to adapt existing models to the statistical requirements of various types of research and has the potential to move these automatic labeling approaches from being unusably inaccurate to being an invaluable new tool. After validating the approach in a number of ways, we annotate the reproductive state of a large dataset of over 600,000 herbarium specimens. The analysis of the results points at under-investigated correlations as well as general alignment with known trends. By sharing this new dataset alongside this work, we want to allow

**Data availability statement:** We share along with this paper two sets of data: first the annotated 600k specimens dataset, with the Budding, Flowering, Fruiting and Reproductive annotations and corresponding confidences. We also share the aggregated species-level flowering season shift used to generate the Flowering shift analysis results, the processed dataset from the phylogenetic signal analyses, and the raw data used to generate the phylogenetic tree figure. The data can be obtained at Zenodo under the following DOI: https://doi.org/10.5281/zenodo.14056635. Code samples allowing to generate an accuracy/coverage graph from a toy dataset, as well as the full code reproducing the 600k dataset large-scale flowering shift analysis are available at https://github.com/NouFuS/confident_based_deep_learning_for_ecology.

**Funding:** This work was supported by the Yale Institute for Biospheric Studies through an G. Evelyn Hutchinson Environmental Postdoctoral Fellowship (this funded the salary and research expenses of QB). The funders had no role in study design, data collection and analysis, decision to publish, or preparation of the manuscript.

**Competing interests:** The authors have declared that no competing interests exist.

biologists to gather insights for their own research questions, at their chosen point of accuracy/coverage trade-off.

---

## Author summary

In recent years, museums and research institutions have digitized vast collections of plant and animal specimens, creating an enormous amount of images and data. Scientists are eager to enhance this data by adding more detailed traits, but manually labeling each specimen is time-consuming and expensive. Deep learning methods, such as convolutional neural networks (CNNs), offer a way to speed up this process by automating labeling. However, these methods are often not accurate enough for scientific use, typically achieving only 80-85% accuracy. In this study, we introduce a method to significantly improve the accuracy of automatic labeling. By assessing the "confidence" of the model's predictions and setting a threshold to filter out uncertain labels, we show that accuracy can be increased dramatically. A basic model with 86% accuracy, for example, can exceed 95% accuracy (by rejecting 40% of labels) or even 99% accuracy (by rejecting 65%). This allows researchers to adjust the trade-off between accuracy and data coverage based on their needs.

To demonstrate the method's usefulness, we applied it to a dataset of over 600,000 herbarium specimens, specifically labeling their reproductive state. This new dataset, shared alongside our study, provides biologists with a valuable resource to explore patterns in plant reproduction with confidence.

## Introduction

Data generation within ecology and related fields is more rapid than ever as automatic data collection technologies ranging from citizen science to camera traps to satellites has proliferated. At the same time, there are large-scale efforts to digitize existing data sources such as natural history collections. However, while the abundance of this raw data is exploding, the time and effort involved in cleaning and annotating this data to make it useful for research can be cost prohibitive.

The need for novel solutions to address the challenges of digitizing and annotating trait data has driven the interest in the potential of AI as is extensively described in [1–3]. In particular, deep learning has been applied in a range of applications, including:

- Detection and/or identification of species or groups of species in images [4,5].
- Classification of particular properties, such as the phenological stages of plants [6] or leaf diseases [7,8]
- At the population level, by aggregating detection and classification results, tools have been developed to provide automated counting and monitoring of wild animal populations, for example with camera trap images [9–12]
- Lastly on an ecosystem scale, satellite imagery can provide estimates of carbon stocks in biomass and track their trends [13].

While field sensors and satellites augment the capacity to monitor ecosystems at an increasingly high frequency and resolution, researchers often rely on historical records to gain insights into long-term patterns. For plants, physical samples are aggregated in herbaria, pressed and dried and mounted on sheets (an example is displayed in Fig 1). These are an important source of information, not only because of the evolutionary and biogeographic diversity of samples, but also because of the temporal distribution of these samples. Many collections possess samples from species that have been collected periodically over the last 200 years, or longer. This temporal component is crucial when performing studies on subjects such as environmental changes, climate change effects, trends of ecological systems evolution, etc.

The physical nature of herbarium samples can make them difficult to access and share among institutions, thus motivating efforts to digitize these collections [14]. Tens of millions of digitized specimens are now available online to researchers across the world through platforms such as the Global Biodiversity Information Facility (GBIF, [15]) and iDigBio [16]. As the digitization bottleneck lessened, labeling became the new bottleneck to performing specimen-based research, as researchers wanted more information than the taxonomic identification, collecting location, and collection date that is typically provided with the samples. Extracting additional information such as visual characteristics from specimens images is usually done manually, either by trained experts, or through crowd-sourcing with consensus techniques.

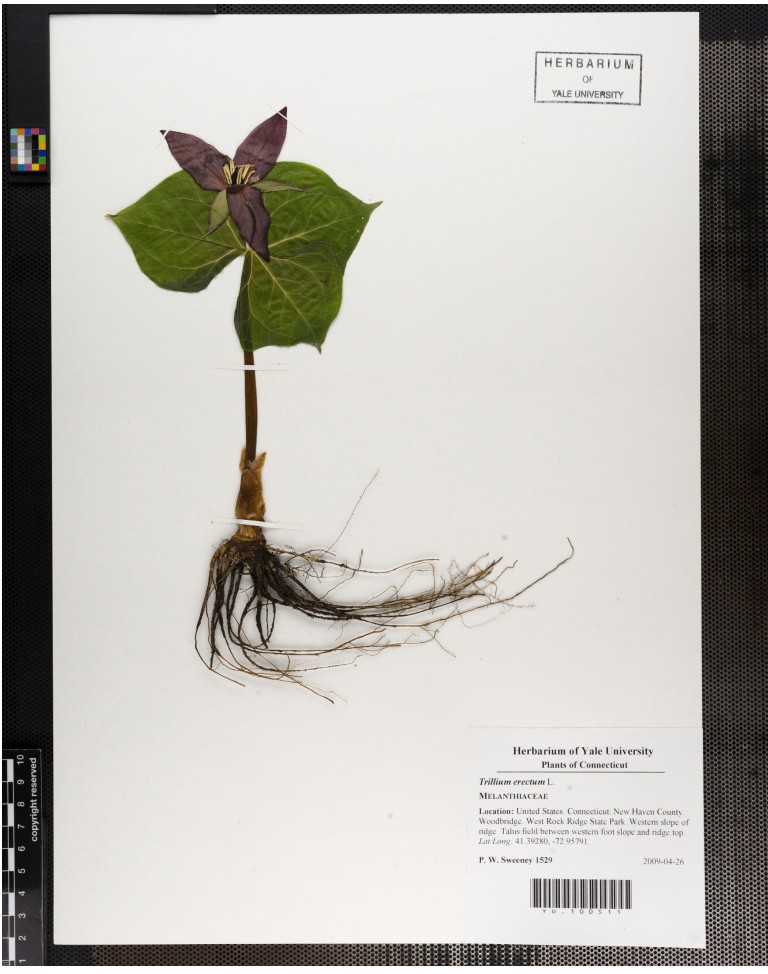

**Fig 1**. **A digital image of an herbarium specimen.** The plant and original label containing the original metadata information, as well as additional reference information for the digital image: a ruler for scale and a color reference grid.

Since a single study often requires thousands of data-points, manual labeling is very labor and cost intensive. Several papers investigate and compare strategies of manual annotations processes to try to improve the efficiency of this step, such as [17] which shows how binary annotation can be a good trade-off against more granular annotations. Other studies have evaluated the difficulty of estimating various characteristics and their expected accuracy [18].

Human-level accuracy for binary phenophase annotation (for example: fruiting or not, flowering or not, etc.) is reported by [18] to be on the order of 95-98%. Recent papers such as [6] have proposed deep learning models to annotate fine-grained phenological features automatically and obtained promising results. These models reported human-level accuracy for coarse categories such as overall fertility but fell short for finer-grained phenophases (e.g. fruiting or flowering) with accuracy of only 80-87%.

Although this type of work shows that applying deep learning models appears to be feasible in principle, the gap between model and human accuracy can prove too substantial to allow researchers to directly use these automatically generated labels, limiting the adoption of this set of automation techniques. An answer to that issue is to use confidence-based methods to increase the trust in the resulting annotations, usually through rejection mechanisms.

The study of classification rejection mechanisms for accuracy-critical applications has been a research topic within the machine learning community for decades, with works such as [19,20] and have been applied to a variety of traditional machine learning classifiers. Lately, several works have been focused on researching ways to get similar results on more modern deep learning classifier models, by learning secondary networks [21] or by modifying the training process to obtain more reliable confidence estimates [22]. Although interesting, these approaches are technically challenging to implement, limiting their adoption by other fields.

In this work, we propose an accuracy/coverage trade-off pipeline that provides an easily implementable and interpretable view of the performance of any softmax-based classifier. We use a reduced version of the risk/coverage analysis introduced in [23] and produce a variation on the traditional risk/coverage curves [20] to make them easier to interpret, notably when trying to link a given confidence threshold to a set of accuracy/coverage values. This can inform the usability of a given model without having to integrate and test it based on its top-1 or top-5 scores alone.

## Confidence-based classification pipeline overview

Traditionally, deep neural classifiers are composed of a set of feature layers that extract information from the input (image in our case) and feed this information into a layer performing a multi-class logistic regression through a softmax layer to output a set of probabilities (one for each class to be predicted). During training, the loss function (training criterion to be optimized) is calculated by comparing the predicted probabilities to the ground-truth class labels (0s and 1s). At inference time, a baseline approach is to select the class with the highest probability as the resulting classification. From there, two standard approaches are to either use the predicted class directly, or to apply thresholding. A typical use of this threshold in the context of binary models is to find an application-specific trade-off between the false-negative and false-positive rates. In this class of approaches, we find the precision-recall analyses and the use of ROC curves plots (Receiver Operating Characteristic). This allows to plot how false-positive and true-positive relates for various threshold values. It is important to note that these metrics are informative only if all the samples are annotated and evaluated against the ground-truth.

For our application, we want to maximize is the *overall accuracy* of the model. To achieve that goal, we will consider a prediction *rejected* if the confidence of the model is below a certain threshold. This process is illustrated by Fig 2 Setting a threshold becomes a trade-off between the overall accuracy of our classifier and how much of the data we will discard (e.g. reducing the coverage). In other terms, we can choose to have a smaller but more accurate set of annotations. By design, this means that we do not measure recall in the usual way (recall would drop as we reject more cases). Instead, we explicitly measure how accuracy improves when uncertain cases are filtered out. Although we could interpret our thresholding as indirectly trading off recall for precision, the key difference is that we evaluate performance *conditional*

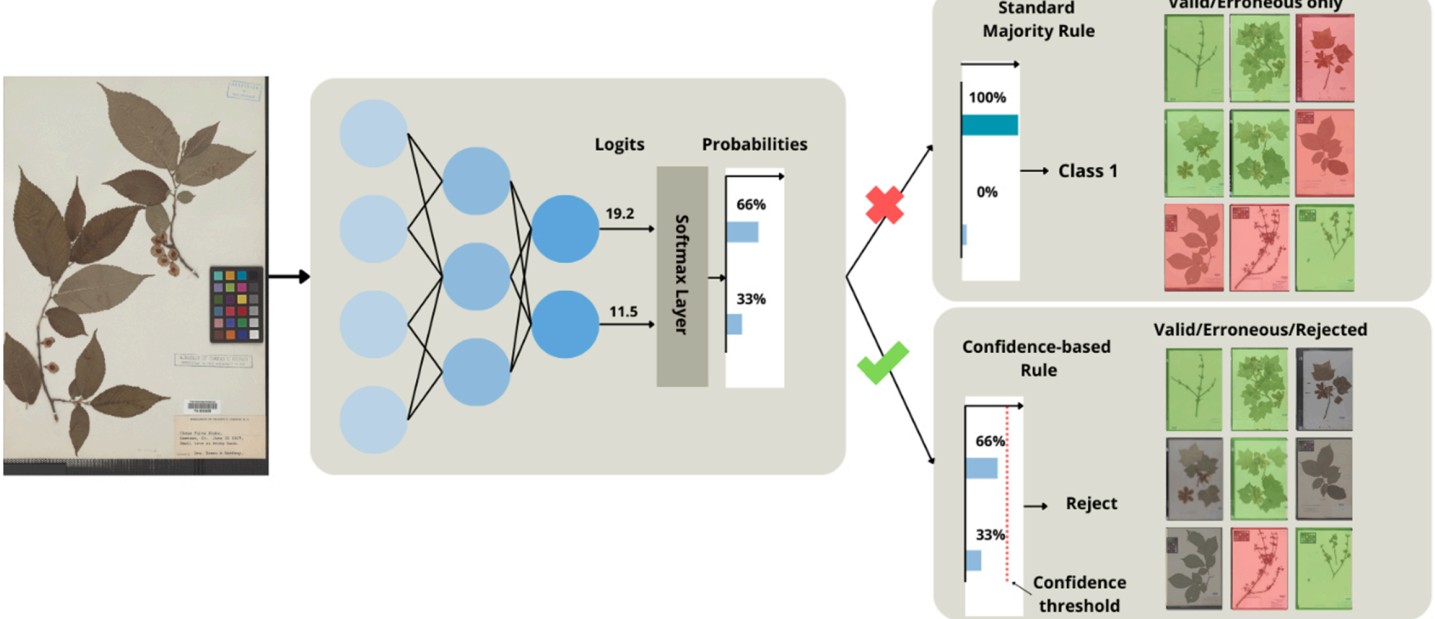

**Fig 2. Overview of the confidence-based workflow.** By only considering labels over a certain probability threshold, we increase the final accuracy of the model at the cost of coverage on the overall dataset (red: wrong label, green: true label, gray: rejected label).

*on acceptance*. In other words, we care about the correctness of the model, since it chooses to make a prediction. This is why we did not compute ROC/AUC or precision-recall curves: those metrics assume decisions on all samples and are less informative when the classifier can abstain.

It is important to note that the model probabilities are not unbiased and are usually only usable on in-distribution inputs (e.g. similar enough to the training data), as too much novelty can lead to classification errors and erroneous probabilities.

We propose a pipeline for end-users of classification models that is easy to implement and to apply to accuracy-critical applications.

When evaluating a model for an accuracy-critical application, the usually reported top-1 or top-5 accuracy may not be enough to predict if the model will have satisfactory performances when replacing a human component with a higher baseline accuracy. In scenarios where unlabeled data is abundant and labeling is costly, solutions without complete coverage provide substantial value. Using a pre-trained model and a validation dataset, we are able to generate accuracy/coverage curves as a function of the confidence threshold which provides a complete overview of model performance under all coverage scenarios. These curves are generated by testing sampled confidence thresholds on the validation dataset and measuring the coverage, (1 - % of rejected predictions) and the accuracy of the *confident* predictions. Figs 3, 7 illustrate the trade-off between accuracy and rejection rates in an easily interpretable manner for non-ML practitioners. Importantly, both of these metrics are generally monotonic with respect to the confidence threshold allowing for an arbitrary target accuracy or a minimum coverage to be selected. From these curves, it is easy to identify which threshold corresponds to a suitable set of accuracy/coverage values. It is then possible to obtain annotations with an arbitrary level of accuracy, specifically in applications where the objective is to employ machine-generated annotations in lieu of human annotations, provided that there is enough data to make up for the reduced coverage.

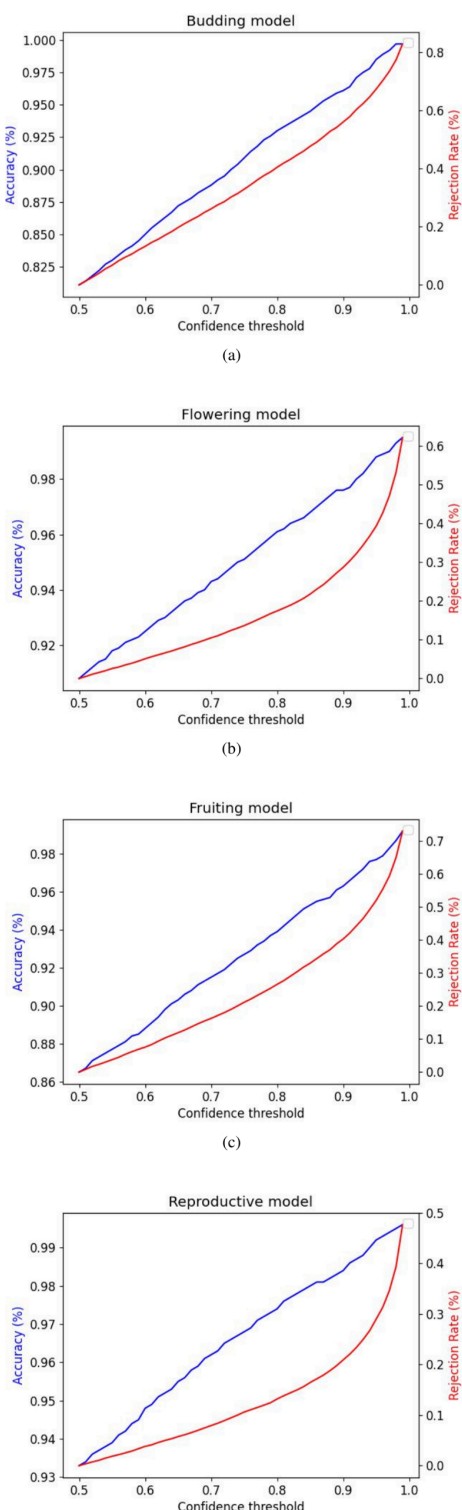

**Fig 3. Rejection/Accuracy curves for all four models.** Associated labels: (a) Budding, (b) Flowering, (c) Fruiting, (d) Reproductive.

# Results and discussion

Through a set of experiments on custom-trained models, as well as on-the-shelf models, we show that it is possible to determine the appropriate accuracy/coverage for a given application. Once that application-dependent constraint is identified, we show that even seemingly subpar classification models can succeed in providing research-grade data. Finally, we apply the described process to annotate a novel New England dataset comprising of over 600,000 digitized herbarium specimens representing around 4000 species in order to analyse flowering season shift at a macrophenological level [24]. After performing an accuracy analysis to determine the appropriate confidence threshold for the task, we perform a preliminary analysis on the influence of several characteristics such as life-form, environment and seasonality on flowering time and find results largely in agreement with notable non-ML studies such as [25–33]. We also present some new findings that could lead to new investigations, as there seems to be an under-researched relationship between the wetland status of the plant environment and its response in flowering season shift. This annotated dataset is shared along with this paper to allow other researchers to gain insights into their own research questions at a larger scale.

## Using the confidence threshold as a mean to reject uncertain samples and increase overall accuracy

By applying the accuracy/coverage trade-off to a custom state-of-the-art binary model (details related to training the models as well as more in-depth insights of the underlying mechanism through embedding-analysis can be found in section Custom phenology models definition and training), we show that it becomes possible to reach an accuracy comparable to human-level. ([18] found that experimented volunteer annotators can reach up to 97% accuracy on annotations (with present/not-present characteristics), at the cost of reducing the annotation coverage down to 30% (on average on our custom models). Choosing a particular threshold will be closely tied to the application constraints and the number of samples available.

For herbarium-based studies, there could be several ways to use this confidence-based method. A straightforward way could be to simply ignore samples that do not meet the confidence criterion. This can be suitable for scenarios where the number of samples available is large enough to be able to work with just a fraction of the initial sample set and retain good statistical quality (as we assume for the study replication in the next section). On the other hand, if the number of samples is critical (*i.e.* high coverage is required), this method can still be used to reduce human effort by automatically annotating a significant fraction of the data and then manually annotating the remaining uncertain samples.

## Replication of a study based on human annotations by exploiting the accuracy/coverage trade-off

We found that using our confidence-based approach, machine-generated labels can be used to eliminate the manual annotation step while obtaining comparable results and conclusions from a manually-labeled phenological study.

The study [34] we replicated compares temporal fruiting patterns of native and invasive New England species. First, a set of digitized herbarium samples was collected and annotated according to their fruiting status. From this, a mean Day of Year (DoY) for fruiting behavior was estimated for each species (with an estimated standard deviation). The study then performs statistical analyses to rule out potential confounding variables, such as temporality, climate, and geography, that could affect the timing of this fruiting behavior. Finally, a statistically significant difference of 26 days is identified between the sets of native and invasive species, the latter having both a later mean DoY and higher variance in fruiting days.

Since this study reports the results of their manual analysis on a per species basis, which is the labeled data that the entire study relies upon, we can treat their results as a ground-truth so we can assess the performance of our automatic labeling in a realistic setting, as well as comparing the final conclusion from our data only. Details concerning the data processing can be found in section Study replication process.

We first look at the impact of the confidence threshold on the mean replication error as well as on the amount of data that is rejected (coverage). Fig 4 shows the evolution of both of these values (in blue the global DoY error between the study and model values, in red the amount of species DoYs that cannot be computed as no samples annotations have

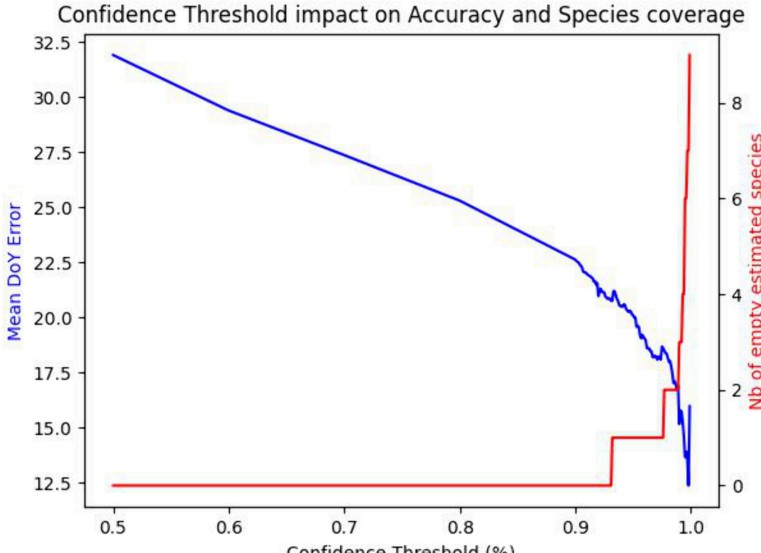

**Fig 4**. DoY estimation error and number of empty species estimates as a function of the confidence threshold.

been accepted). With a baseline model (implicitly using a 50% confidence threshold), the replication error between the mean fruiting DoY over all species is around 31 days. When the minimum confidence threshold increases, the mean fruiting DoY error decreases as the rejection rate and accuracy increase. Eventually, the error rises again when the number of samples becomes so small as to be statistically meaningless. This check against the study data confirms that higher model accuracy translates into a more reliable estimate of the species fruiting DoYs.

The mean error score does not tell by itself if the data generated by the model can be trusted or not. To gain a better insight, we compared the results on a per species basis, as shown on Fig 5 and Fig 6. We start, again, by looking at the "naive" approach, where we use the standard 50% threshold to decide if a samples is fruiting or not. By examining the species per species error displayed on Fig 5, we can see that the mean estimates are nearly always outside of the standard deviation reported by the study, suggesting a mediocre replication and possibly unusable data. When calculating the difference in fruiting DoY between native and invasive species, we get a difference of 9 days in fruiting behaviour, which is not consistent with the 26 days difference reported in the original study.

Next, we apply the thresholding technique presented earlier and select the threshold corresponding to human-level accuracy, which is reported above 97% by [18]. By referring to the accuracy/coverage graph of our fruiting model in Fig 3(c), we can identify the corresponding threshold to be around 0.99, and expect a rejection rate around 72%. The decrease in coverage resulted in three of the species not having enough samples left to estimate their species' mean DoYs, but on the other hand, the overall replication error drops to 14 days, a 50% decrease compared to the error with the naive approach. By comparing on a per species basis as earlier, we can see in Fig 6 that the new estimates are better aligned with the study. For a majority of species, the estimated mean DoYs now lie within the standard deviations reported in the original study suggesting our methodology improves the reliability of the labels. We confirm that by computing the native and invasive species DoY difference and we obtain a result of 25 days, a single day off from the study's estimate of 26 days.

By recreating this study, we show that by taking advantage of the accuracy/coverage trade-off we are able to generate automatic labels which appear sufficiently reliable to be used in research. Furthermore, we also demonstrated that the lower accuracy levels of the base model's labels would lead to erroneous results and conclusions.

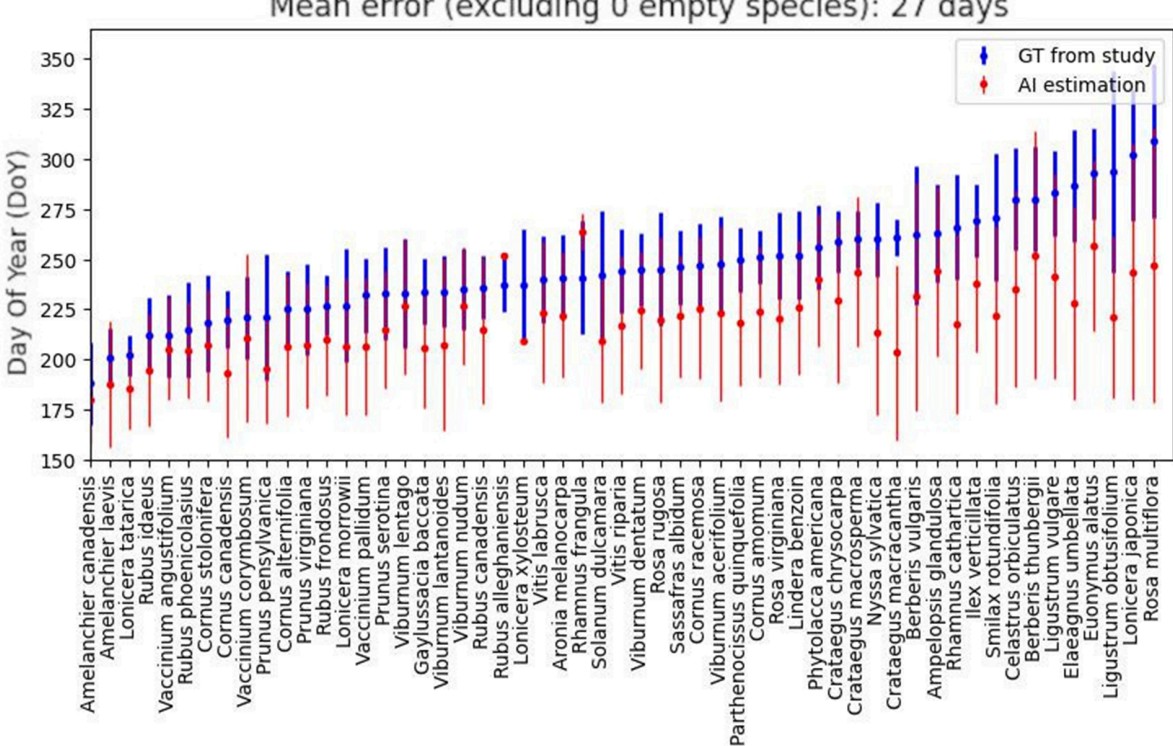

**Fig 5**. **Comparison of study ground-truth and AI estimate, per species, with a confidence threshold of 0.5.** Blue: study ground truth with mean/std, red: AI estimate with mean/std.

## Generalization outside of herbarium samples: Application on INaturalist2018 dataset

We found that this method can also be applied to publicly available multi-class models and generate similar benefits as the previously studied custom binary classifiers. We applied the method to an off-the-shelf model available online [35], trained on the INaturalist2018 dataset, and evaluated the impact of the accuracy/coverage trade-off on its performances. Details on the INaturalist2018 dataset and the off-the-shelf model can be found in section INaturalist2018 dataset and classification model.

The main difference compared to the previous experiments is that the classifier here outputs classification probabilities for the 8142 species represented in the dataset (compared to the 2 classes of our binary classifiers). From our evaluation on the validation dataset, it achieves a top-1 accuracy of 44% over the 8,142 categories.

Fig 7 shows the accuracy and rejection rates with respect to the confidence threshold when applying the proposed process to all available data-points in the validation dataset, all categories considered. We can see that the two curves follow the same trends as the ones presented earlier on herbarium samples, although the classifier is now multi-class.

The impact on accuracy is more pronounced than before, since we are able to go from an accuracy of 43% up to more than 90% (with 82% rejection rate). Sampled points from the graph are presented on Table 1 for clarity.

This experiment highlights that the process performs in the same way with an off-the-shelf multi-class model, trained on a publicly available dataset as it did on our custom-trained models. This suggests that the proposed method could be applied to a large variety of classifiers, across different classifier model architectures and different types of images contents.

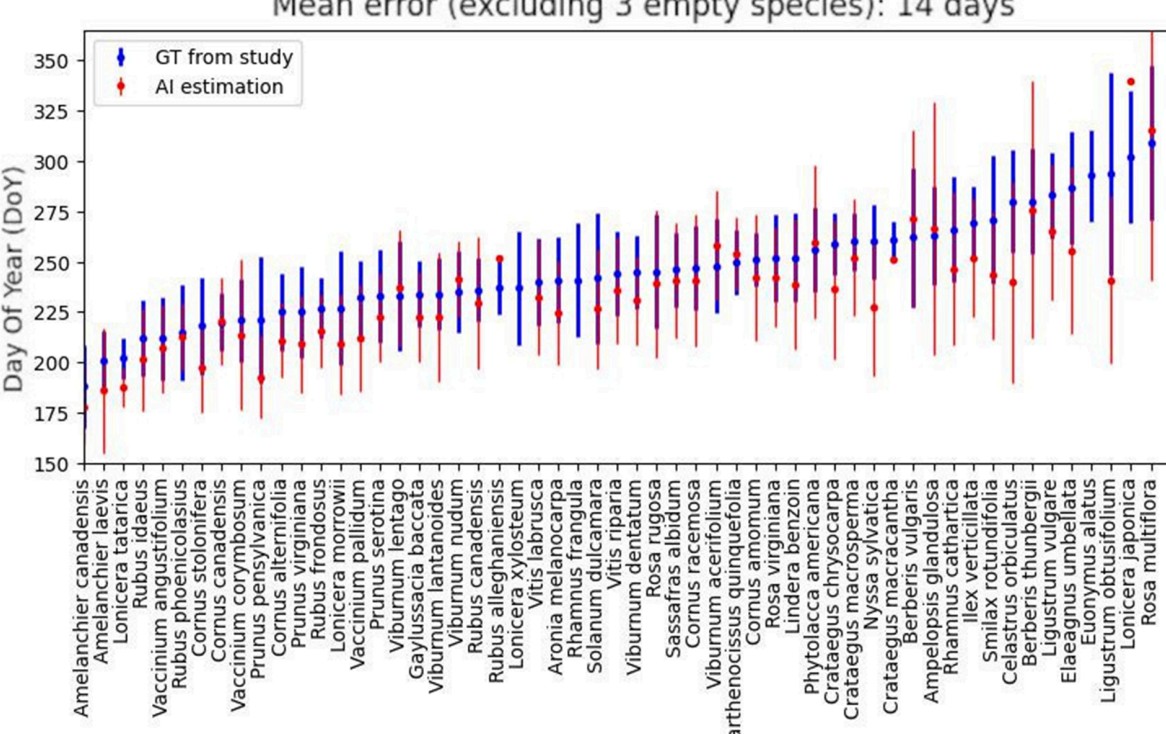

**Fig 6**. **Comparison of study ground-truth and AI estimate, per species, with a confidence threshold of 0.99.** Blue: study ground truth with mean/std, red: AI estimate with mean/std.

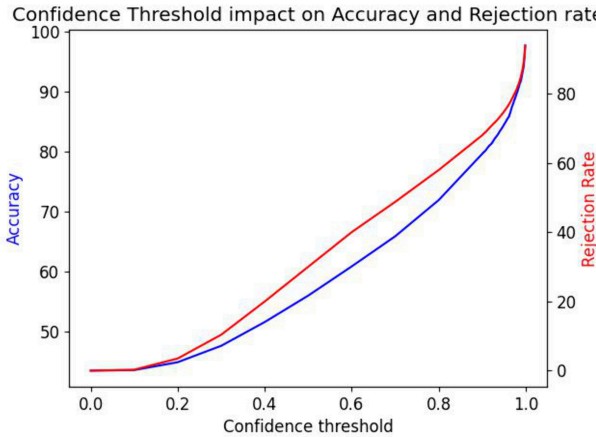

**Fig 7**. **Rejection/Accuracy curves of the INaturalist2018 multi-class classifier.**

## Investigating flowering time shifts using a novel 600K specimens dataset

Over the past couple of decades, there have been an increasing number of studies reporting on plant reproductive phenological change at various scales [25,33,36–42], all motivated by exploring the impacts of anthropogenic climate change

**Table 1**. **Expended accuracy results on validation set of lNaturalist2018 with confidence thresholds.** Accuracy and rejection rate percentages for particular minimum confidence thresholds.

| Min. confidence | 0% | | 50% | | 90% | | 99% | |
|---|---|---|---|---|---|---|---|---|
| Acc./Rejection % | | | | | | | | |
| All Cat. | 44 | 0 | 56 | 30 | 80 | 58 | 93 | 85 |

on phenology. The growing availability of new tools and large datasets makes it increasingly possible to examine phenological patterns across larger taxonomic, geographic, and temporal scales ("macrophenology" sensu [24], see [41] for a recent example).

Here we explore the macrophenological potential of an automatically labeled 600K dataset by examining phenological changes for a large number of species at regional scale and over a many-decadal time span. For each species, we evaluate the change in flowering time using linear regression, yielding for each species the direction, magnitude, and statistical significance of change. We then calculate the average shift over all species. Fig 8 illustrates visually the workflow that is described in this section. Details on the dataset definition, annotation and processing can be found in section 600k Dataset definition, annotation and analysis.

**Overall analysis results.** It is important to account for the influence of phylogeny when investigating the drivers of phenological shifts [43–45]. Our tests for phylogenetic signal detected no significant signal for species showing either shifts to earlier flowering or later flowering, though several of the traits analyzed in our subset analyses did show strong

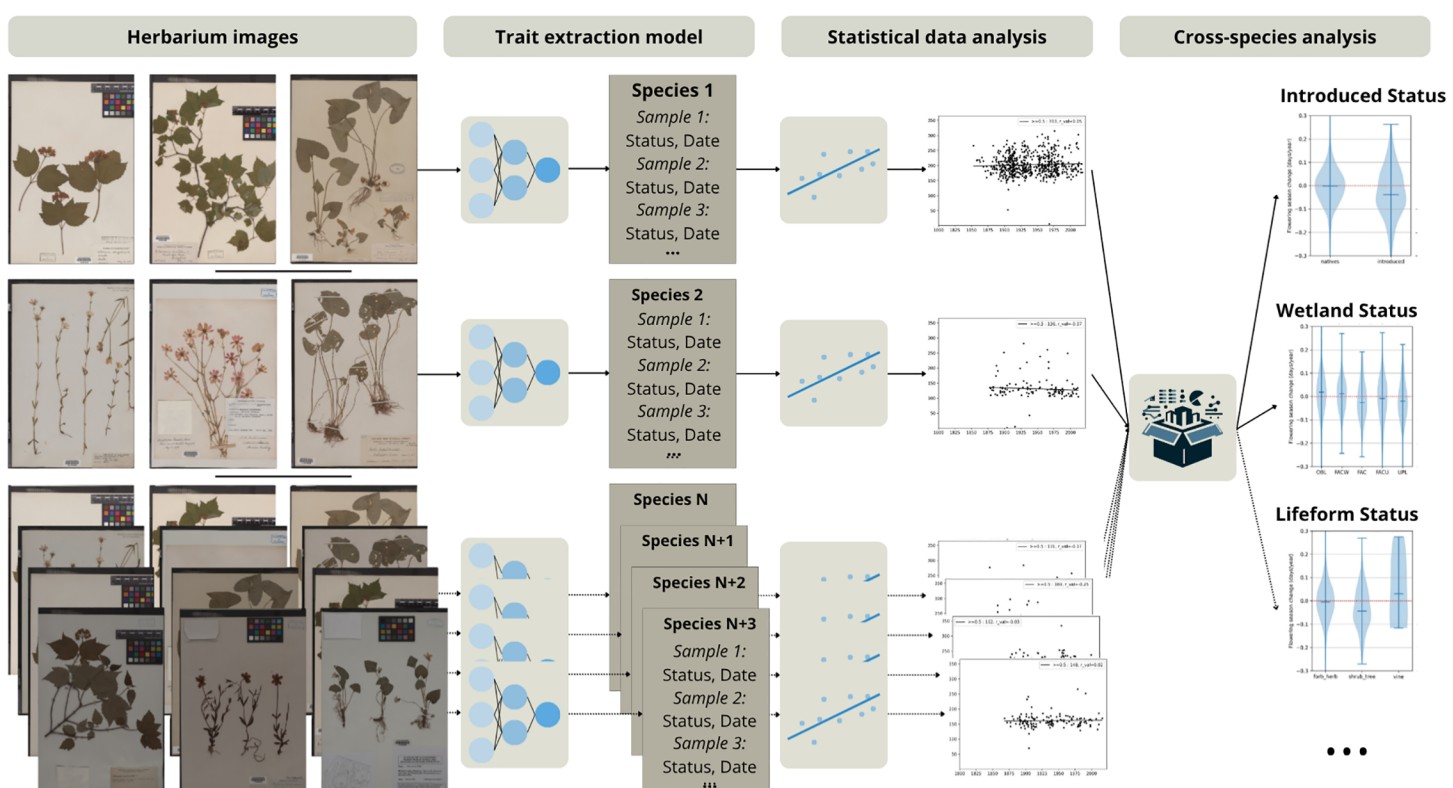

**Fig 8**. **Workflow to automatically obtain a macrophenological analysis of flowering time shifts: from non-annotated samples to regional trends.**

phylogenetic signal (native vs invasive, lambda= 0.744; woody vs herbaceous, lambda=1.00; flowering seasonality, lambda=0.878). Given this result, we proceeded with analyzing the full 680 species dataset.

When analyzing the full 680 species dataset, 176 species had significant ($p < 0.05$) shifts in flowering time. Of these, 102 of these species showed a shift to an earlier flowering time and 74 shifted to a later flowering time. The average shift was –0.248 days/decade (-0.0248 days/year). The greatest negative shift was –3.78 days/decade (in *Crepis capillaris* (L.) Wallr.) and the greatest positive shift was 4.33 days/decade (in *Callitriche palustris* L.). 504 species exhibited no significant shift in flowering time.

Other studies have examined flowering time shifts in plants from the geographic region targeted in our study. Our results are comparable to, although less than, the –0.44 to –0.68 days/decade average shifts reported at other New England localities ranging up to 161 years during the last two centuries [25–29]. Other studies, while showing that species shifted their flowering times in response to warmer temperatures, did not find a net shift in flowering time over time [38,46].

**Subset analyses.** Many studies have noted the importance of species-specific plant traits and characteristics as important drivers of phenological response [33,47–50]. To further demonstrate and explore the utility of our large phenological dataset, we analyze the impact of five plant characteristics on flowering time shifts: growth form, nativity status, wetland status, seasonal timing of flowering, and flowering duration.

Without taking phylogeny into account, several of our examined characteristics were potential drivers of phenological response. Visualizations of our results for our trait-focused analyses can be found in Figs 9, 10, 11, 12, 13 and significant results are summarized in Table 2.

When examined within an evolutionary framework, most of characteristics that we suspected might influence phenological response did not show any correlation with either significant shifts to earlier or later flowering, with one exception. The seasonality of flowering (whether a species flowers in the first or second half of the growing season) strongly predicted whether and how species demonstrated a significant shift in flowering: early-season flowering species were more likely to shift toward earlier flowering times (p = 0.0001, Table 3), and late-season species were more likely to shift toward later flowering times (p = 0.01, Table 3). We provide a visualization of the flowering seasonality character and the detected flowering shift for each species on a phylogenetic tree: Fig 14.

**Noteworthy findings from flowering time shift analyses.** Our analyses demonstrate the value of ML approaches to macrophenological studies. While mass digitization activities have generated huge numbers of specimen images

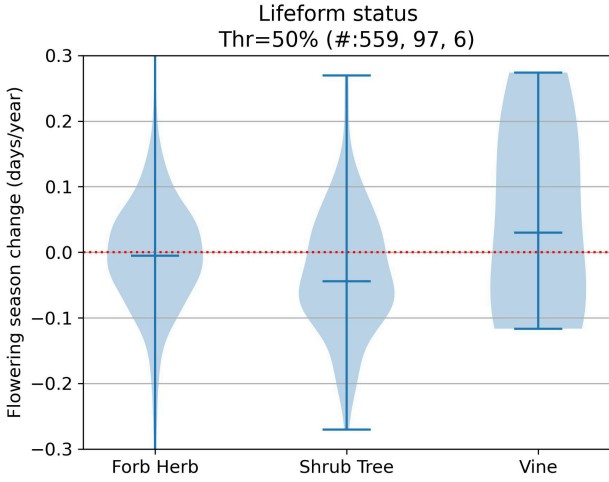

**Fig 9**. **Flowering shifts distribution per growth form status.**

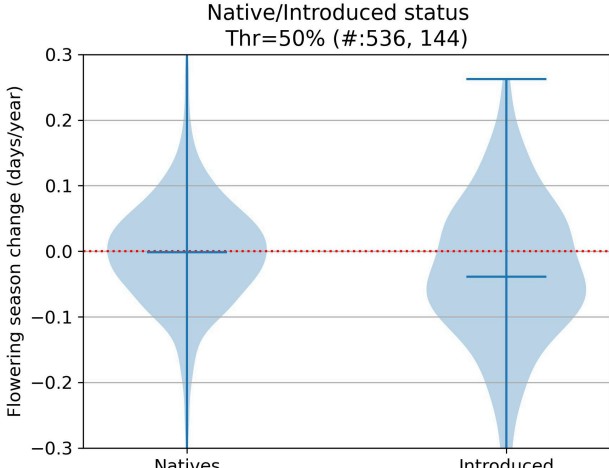

**Fig 10**. **Flowering shifts distribution per native/introduced status.**

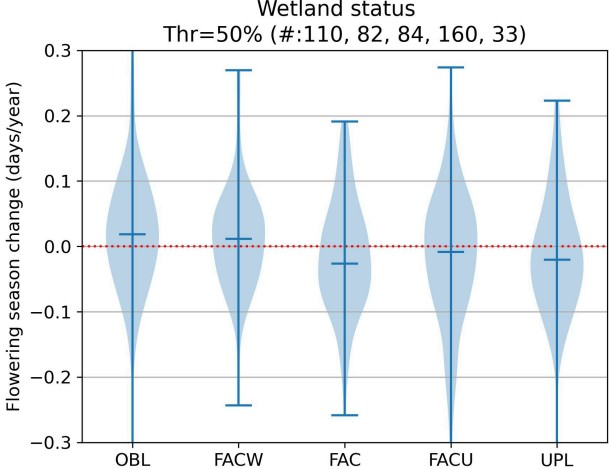

**Fig 11**. **Flowering shifts distribution per wetland status.**

and occurrence data, it has created a new challenge of how to extract trait data that is represented in the specimens. The approach outlined here demonstrates the power of ML methods to address this challenge. We were able to analyze flowering time shifts in almost 700 species, examining the differences in shifts among various plant characteristics. For the species we examined we show a significant 0.248 days/decade average shift to earlier flowering times over a many-decadal time span during the last two centuries.

Our results were largely consistent with other phenological studies examining flowering time shifts for fewer numbers of species. In agreement with other studies, we also show that growth habit, nativity status, and seasonal time of flowering significantly influence shifts in flowering time (e.g., growth form: [47,51]; nativity: [32,33]; seasonal timing of flowering: [28,29]). Unlike some studies [28], we found no significant difference in flowering time shifts between species with narrow versus broad flowering time duration.

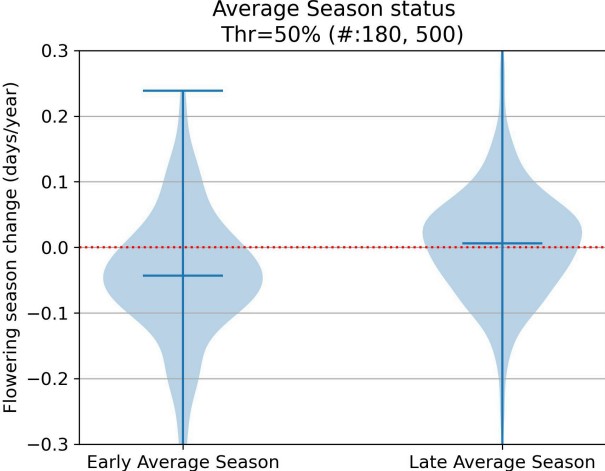

**Fig 12. Flowering shifts distribution per early/late season status.**

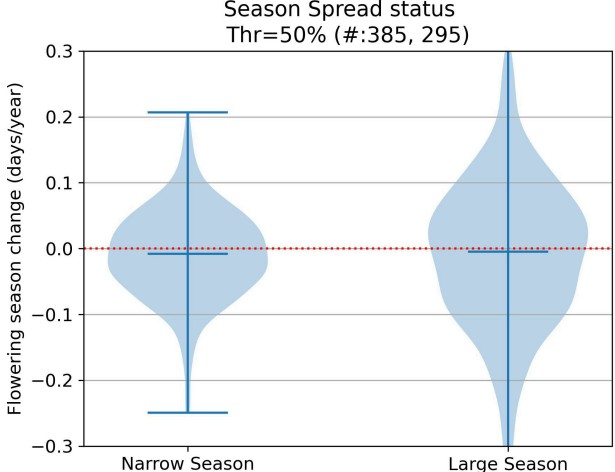

**Fig 13. Flowering shifts distribution per season spread status.**

**Table 2. Results of Welch's T-test evaluating effect of plant characteristics on flowering time shift.**

| Characteristic | Comparison | Direction of response | Magnitude (days/year) | p value |
|---|---|---|---|---|
| Growth form | shrub/tree vs forb/herb | earlier | 0.025 | 0.014 |
| Nativity | native vs non-natives | earlier | 0.03 | 0.003 |
| Wetland status | FAC vs OBL species | earlier | 0.036 | 0.006 |
| Wetland status | FACU vs OBL species | earlier | 0.036 | 0.003 |
| Wetland status | FAC vs FACW species | earlier | 0.026 | 0.048 |
| Wetland status | FACU vs FACW species | earlier | 0.026 | 0.035 |
| Seasonal timing of flowering | Earlier season vs late season | earlier | 0.046 | 3.45e-08 |

A novel result of this study is our finding that species that have a higher affinity for wetter habitats shift their phenology relatively less than those in dryer habitats. We are not aware of other studies that make direct comparisons of influence of wetland status on phenological shifts. The underlying driver for these differences is not obvious.

**Table 3**. **Results of Pagel's test for correlated evolution between two binary characters.**

| Character | | log-likelihood of model fit | | AIC of model fit | | p value |
|---|---|---|---|---|---|---|
| dependent variable | independent variable | dependent model | independent model | dependent model | independent model | |
| significant shift | native/introduced | –587.7941 | –588.1085 | 1187.588 | 1184.217 | 0.73 |
| significant shift | flowering season range | –696.8083 | –696.8406 | 1405.617 | 1401.681 | 0.968 |
| significant shift | flowering seasonality | –604.293 | –606.742 | 1220.586 | 1221.484 | 0.086 |
| significant shift | growth form | –436.4495 | –434.7207 | 884.8989 | 877.4415 | 1 |
| significant early shift | native/introduced | –496.7366 | –498.821 | 1005.473 | 1005.642 | 0.124 |
| significant early shift | flowering season range | –607.5528 | –607.6502 | 1227.106 | 1223.3 | 0.907 |
| significant early shift | flowering seasonality | –508.6071 | –517.4496 | 1029.214 | 1042.899 | 0.0001 |
| significant early shift | growth form | –344.4841 | –346.8294 | 700.9682 | 701.6588 | 0.096 |
| significant late shift | native/introduced | –460.2825 | –460.8659 | 932.565 | 929.732 | 0.55 |
| significant late shift | flowering season range | –569.548 | –569.596 | 1151.096 | 1147.192 | 0.953 |
| significant late shift | flowering seasonality | –474.899 | –479.496 | 961.798 | 966.991 | 0.01 |
| significant late shift | growth form | –307.2972 | –307.2581 | 626.594 | 622.516 | 1 |

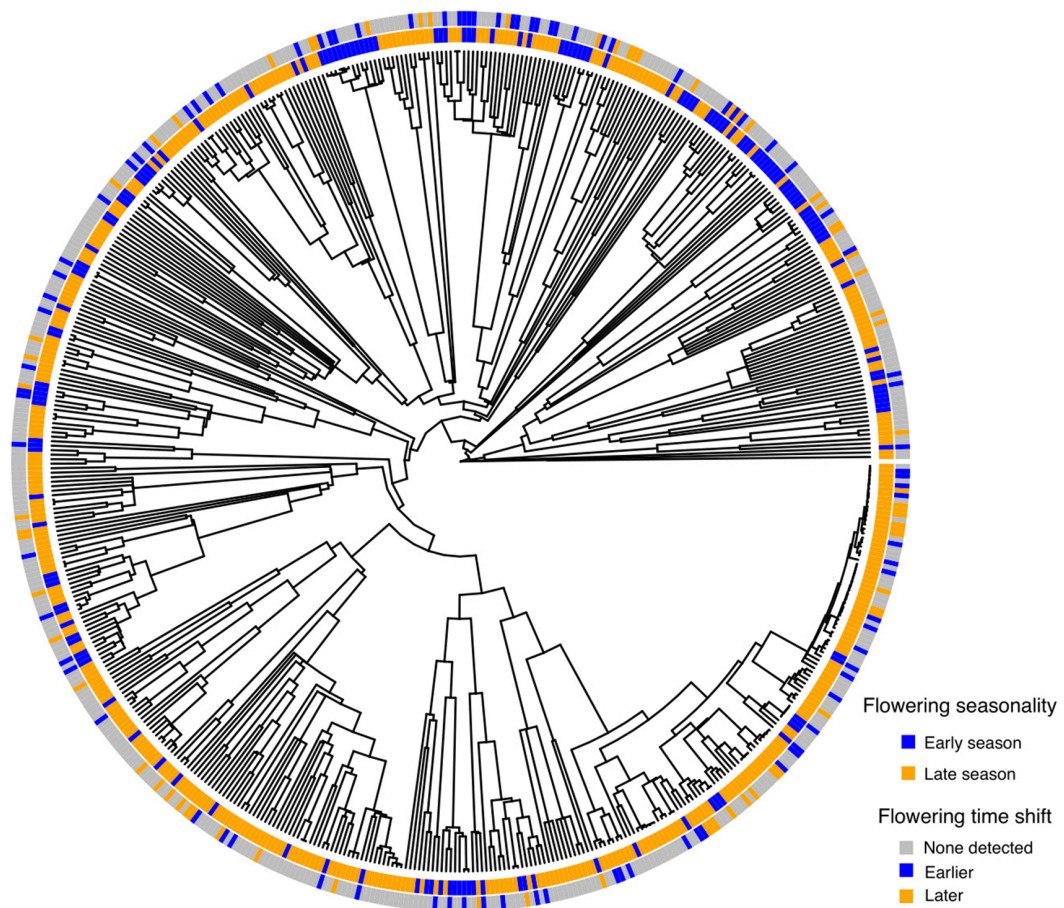

**Fig 14**. **Phylogenetic tree with seasonality character and detected flowering shifts.**

Another important result of our study concerns the influence of seasonal timing of flowering on shifts in flowering time. Our analyses show that species that flower earlier in the year have shifted their flowering times earlier and those that flower later in the season have shifted their flowering times later. Many studies have shown that spring flowering species

have shifted their flowering times earlier (e.g., [26,28,29,33,52,53]), but fewer have shown that in addition, late season flowering species shift their times later [41,54,55]. Our macrophenological study in one of first suggest that this phenomenon is operating at a regional scale and affecting many species. This result is in line with non-phenological studies that show that the growing season, defined as frost-free days, is lengthening in North America [56]. This has important implications plant-pollinator interactions, species competition, and ecosystem functioning (e.g., [57–61]).

We found that there is considerable variation among species in flowering time shifts, some relating to life history characteristics. This result highlights the need for more detailed investigations at the species and trait level and calls into question the interpretation of average or aggregate values computed across numerous species. Interpretation of such aggregate data is difficult, and requires additional knowledge of the species and organisms being studied. These concerns parallel those that surround interpretation of large-scale phylogenetic studies [62,63]. The influence of other factors, herbarium sampling biases [64,65] or other data quality issues [66], for example, must also be considered. Such issues are particularly challenging when datasets consist of hundreds of thousands to millions of occurrence records. Another issue concerns taking phylogeny into account when studying drivers of phenological change. Some variation in phenological response could be explained, in part, by phylogeny [43–45].

## Conclusion

We presented a practical methodology to apply deep learning techniques to large scale ecological studies. By using a confidence thresholding framework to already trained deep learning models we showed that we can sacrifice a portion of the available samples in order to reach an arbitrarily high annotation accuracy on ecological tasks.

We showed that we are able to replicate in an accurate manner a study that required the manual annotation of 15,000 samples by using only deep learning model annotations generated in a matter of hours on a GPU, from a model that could have been seen as too inaccurate from its top-1 score alone. From these labels, both the species-level phenology temporal patterns as well as the study conclusion proved to be in similar ranges. Finally, we gathered, annotated automatically and analyzed a novel 600,000 herbarium samples dataset to demonstrate the potential of our approach to speed up and increase the scale of studies. The large scale analyses we performed by looking at the flowering shift of various characteristic subsets were mostly in-line with the current state of the art in macrophenology, and further querying along less commonly researched categories brings interesting research paths. We share this annotated dataset alongside this paper as a way to enable ecologists to gather insights for their own research questions from this dataset.

We believe that this approach has a great potential to put deep learning tools more easily in the hands of ecology researchers by allowing practitioners to reach human-level accuracy annotation by reducing the coverage, especially with models that would traditionally have been deemed as too inaccurate in comparison to human annotators. By getting more out of current state-of-the-art models, this could speed up significantly the research cycle, by allowing to obtain preliminary answers to research hypotheses without committing large amounts of precious human resources.

## Materials and methods

### Custom phenology models definition and training

We describe here the training dataset, the model architecture and associated training strategy, including the hyperparameters used and data augmentation strategies, and finally the performance metrics reported on the validation dataset.

**Dataset.** For training our models, we use a custom dataset of 47,551 digitized herbarium sheets, generated primarily as part of the New England Vascular Plants (NEVP) [67] project, that have been annotated by human experts with the following phenological stages: budding, flowering, fruiting, non-reproductive. A single sample can exhibit several characteristics simultaneously (i.e. budding, flowering, and reproductive). Labeled training samples span 40 taxonomic

families whose phenological characteristics can be identified from a high resolution image with the human eye (i.e. excluding species where a microscope would be necessary).

From the manual annotations, we obtained the following statistics for the labels of interest in the training dataset:

- Budding: 13350 samples (28%)
- Flowering: 24369 samples (51%)
- Fruiting: 21460 samples (45%)
- Non-Reproductive: 7386 samples (15%)

From this initial dataset, a standard split is performed by assigning 80% of the samples to a training set, and the remaining 20% to a validation set to evaluate model performances. This dataset is large enough that a standard split into training/validation/testing sets will lead to stable results, without needing to use a more expensive training scheme such as cross-validation or performing several runs and using ensemble voting to account for volatility in the model results. Again, we are also trying to keep the deep learning aspect of the method as straightforward as possible to highlight the gains for non-ML practitioners.

**Network architecture: Xception.** For this work, our goal was to automatically label the four phenological stages. Because each specimen could have multiple labels, we decided to train four separate binary classifiers (one for each category). This design choice follows [17] and allows performance to be compared with [6]. We chose the deep convolutional neural network architecture 'Xception', first introduced in [68]. Due to the drawbacks of training deep neural networks from scratch (both in terms of computing power and quantity of training data required), we employ the standard method of using a model pre-trained on the ImageNet classification dataset, replacing the last layer with a binary classifier (2 classes) and finally training this modified model on the annotated herbarium dataset (fine-tuning).

**Data preprocessing pipeline.** Training a network with a relatively small dataset such as ours can result in a classifier with low generalization capability. Thus, we employed data augmentation, a standard approach to add robustness to the training process and prevent over-fitting. Specific data-augmentation strategies avoid relying excessively on certain color properties, as these can vary greatly depending on the age or initial state of the sample. In this work, we employed the following types of augmentations provided by the PyTorch library after resizing each image to a 299x299 dimension (required input size for an Xception network):

- Color Jitter, with brightness, contrast, and hue factors at 0.2
- Random Grayscale, with a probability of 0.1
- Random Rotation, with a maximum rotation of 90°
- Random Horizontal Flip, with a 50% probability
- Random Equalization, with a 50% probability

**Training process.** We then trained our four binary classifiers, corresponding to each of the labels of interest: budding, flowering, fruiting, non-reproductive. For each network, the class imbalance in the training set was addressed by weighting each sample contribution according to its class. The training was performed with the Adam optimizer, with learning parameters such as $lr = 0.001$, $\beta_1 = 0.9$, $\beta_2 = 0.999$. Models were trained on a V100 GPU, with a batch size of 200. Using an early stopping method based on a loss plateau detection, each training ran for about 30 epochs.

**Results on validation and test sets.** The performance evaluation of the four models on the validation set are reported on Table 4.

These performances are in the same range as reported by [6], confirming the performance limitations for this category of networks when trained from this amount and type of data.

**Table 4**. Accuracy results on validation set.

|                  | Accuracy (%) |
|------------------|--------------|
| **Budding**      | 81.1         |
| **Flowering**    | 86.3         |
| **Fruiting**     | 82.4         |
| **Non-Reproductive** | 91.4     |

These performances correspond to a confidence of over 50% (where no prediction is rejected), we can extend the results to see the effect of various confidence thresholds and the corresponding impact on the overall accuracy and rejection rate (Table 5). More fine-grain results can be seen on Fig 3.

**Underlying mechanism insights: Analysis of embeddings.** To further validate our method and better understand its behavior, we examine the distribution of confidence scores in the embedding space of the trained networks. For the mechanism to be sound, we would want to observe lower confidence scores to be located at the interface between the two classes. This would indicate that rejecting samples with low confidence scores is similar to enforcing a larger decision boundary between classes. To generate that visualization, we extracted the embedding vector of the last hidden layer of the "Flowering" classifier for each point of the validation dataset and obtained a 2D map with the dimensionality reduction algorithm, t-SNE [69]. The result can be seen in Fig 15(a). The two classes are clustered appropriately, as confirmed by the reasonable classification accuracy, but there is a region where the two clusters are in contact, suggesting that class separation in this area is degraded. This region is unsurprising given that there is a continuous transition from budding to flowering but this particular model can only indicate flowering or not flowering. Fig 15(b) represents the confidence score of each point with a color ranging from green (high confidence) to black (low confidence). As expected, the closer to the area where the two clusters border each other in the embedding space, the less confident the estimation. By rejecting low confidence points, we can then effectively separate the clusters, increasing the class separation (or decision boundary) in embedding space, and ultimately, increasing the quality of the classification. Examples are shown in Fig 16 to illustrate the types of specimens resulting in High or Low confidence. High flowering confidence classifications are generally linked to large and visible flowers, e.g. Fig 16(a,b). On the other hand, high non-flowering confidence is usually linked to either the presence of features that exclude the possibility of the class, such as fruits that cannot co-exist with flowers, e.g. Fig 16(c) or the complete absence of any hints of reproductive features, e.g. Fig 16(d). Finally, unsure classifications usually do not possess these explicit characteristics, exhibiting ambiguous small reproductive structures, e.g. Fig 16(e), or small features that could be either unrelated to reproduction but that share visual similarity to flowers to some extent, e.g. Fig 16(f).

**Limitations and potential improvements.**

**Calibration.** For the sake of simplicity, we used the network confidence scores directly. It is important to note that these scores are rarely calibrated directly after training, which means that scores do not directly relate to accuracy. Although they are monotonic, with an increase in confidence leading to an increase in accuracy, the absolute values are not identical (otherwise a classification with 50% confidence would lead to an accuracy of 50%). This class of problem,

**Table 5**. **Extended accuracy results on validation set with confidence thresholds.** Accuracy and rejection rate percentages for particular minimum confidence thresholds.

| Min. confidence | 50% | | 75% | | 90% | | 99% | |
|-----------------|-----|---|-----|----|-----|----|-----|----|
| **Acc./Rejection %** | | | | | | | | |
| Budding         | 81   | 0 | 89.4 | 30 | 95.5 | 52 | 98.3 | 77 |
| Flowering       | 86.3 | 0 | 91.8 | 19 | 95.1 | 37 | 98.6 | 70 |
| Fruiting        | 82.4 | 0 | 88.9 | 23 | 93.6 | 43 | 98   | 77 |
| Non-Reproductive | 91.4 | 0 | 95.0 | 11 | 97  | 21 | 98.6 | 48 |

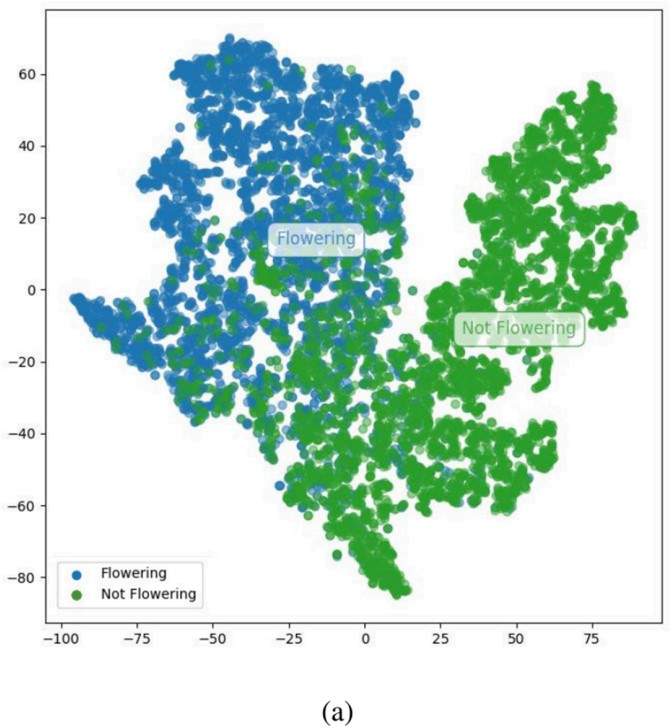

(a)

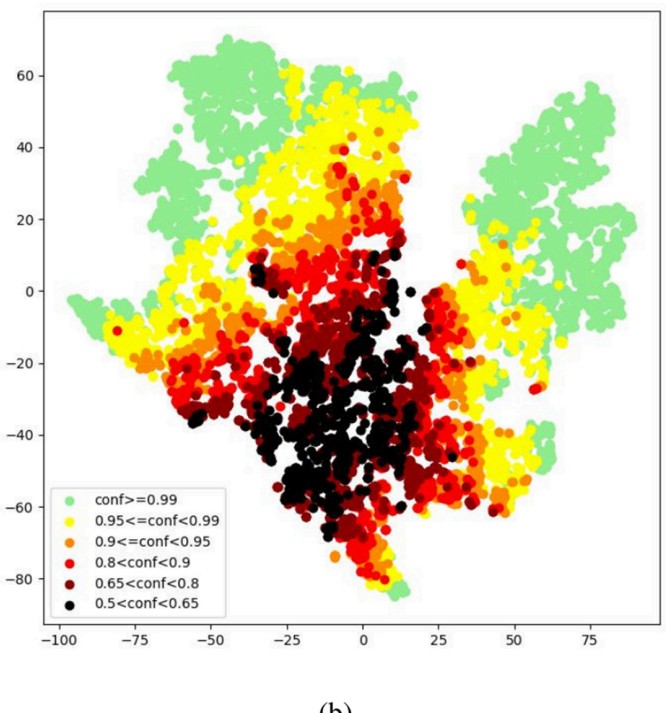

(b)

**Fig 15**. **2D projection of the Flowering classifier's embeddings for the validation dataset from a t-SNE algorithm.** (a) Blue and Green are the Flowering/Not Flowering classes respectively. (b) Additional overlay with the confidence scores.

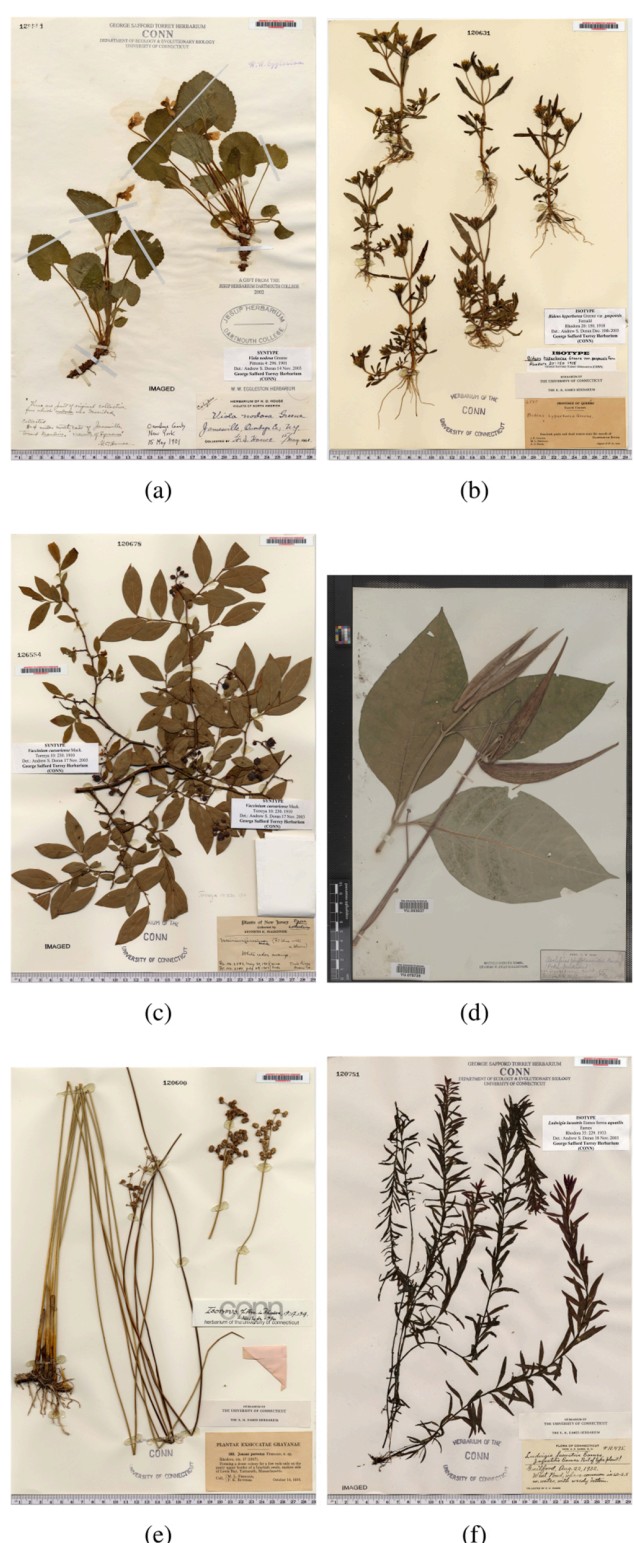

**Fig 16. Examples of images with Sure and Unsure Flowering classification.** (a,b) High Flowering Confidence (a: 97%, b: 94%), (c,d) High Not-Flowering Confidence (a: 98%, b: 99%), (e, f) Unsure Flowering (e: 41%, f: 62%).

referred to as Model Calibration is well studied and numerous methods have been proposed to improve calibration. Methods to learn calibration post-processing strategies can be found for example in [70], both for binary and multi-class classifiers, as well as in [71] for a more in-depth review of designing neural networks with better calibration properties.

Although our rejection/accuracy curves allow one to empirically find the threshold that matches a given set of accuracy and coverage requirements, adding a calibration process to the trained model would make the selection of the threshold more intuitive.

**Sampling bias.** Another concern that can arise while using this type of method is the risk of sampling bias. In fact, by selecting samples with greater confidence, the method will focus on samples that are easy to classify, favoring some species or phenological stages more than others. This means that our proposed method will perform better in some data subsets than in others. A consequence is that our method is better suited to large-scale analysis where this sampling bias will get smoothed out and where losing small data subsets will not affect the final analysis results. In cases where a more uniform coverage is needed, such as maintaining enough data per species for example, this will require either training mode specialized models on harder and more specific datasets, or meshing it with human labeling, using the rejection score of each species as a guide on where to use human time for a higher impact.

### Study replication process

**Data collection.** The study [34] reports having collected 15K samples by extracting all the samples corresponding to a set of 55 species from 6 groups of New England herbaria, namely: the Harvard University Herbaria (A, FH, GH); the George Safford Torrey Herbarium at the University of Connecticut (CONN); the University of Maine Herbarium (MAINE); the Hodgdon Herbarium at the University of New Hampshire (NHA); the Yale University Herbarium (YU); and the University of Massachusetts at Amherst Herbarium (AC, MASS,TUFT).

By querying the Consortium of Northeastern Herbaria portal [72] for the same collections and the same species, we collected a set of roughly 16K samples, which is on the same scale as the amount reported by [34]. We then automatically annotated these samples with the previously described fruiting classifier.

**Data processing.** Since our fruiting classifier only provides a binary label of whether or not the specimen is fruiting (due to the nature of labels of our training set), there are two key differences between our fruiting classifier outputs and the manual annotations of the study.

First, the manual annotators in the study marked fruits looking too mature (according to the sample collection date) to plausibly be from the current growing season and then added 365 days to the day of year (DoY) of the sample since they believed the fruit was from the previous year. To roughly replicate this process, we considered all fruits found in January and February to be from the previous year and added 365 days to the DoY of these samples. Due to similar motivations, the annotators discarded all samples collected between March and May, so the same procedure was followed in our experiment as well.

Second, the annotators in the study were not labeling all fruiting samples equally but focused on *ripe fruits*. We expect this to cause a bias compared to our model (which was trained on labels all fruits regardless of the maturity), shifting the mean DoYs of fruiting earlier in the year. Additional processing could be done to rectify this, such as systematically discarding the earliest fruiting samples labeled by our model. For simplicity, we did not do any additional processing in this study under the hypothesis that this bias affects all species and thus does not change correlations between species substantially.

### INaturalist2018 dataset and classification model

The INaturalist2018 dataset has been created for an online AI competition. It contains 437,513 training images, 24,426 validation images, and 149,394 test images. These images represent a total of 8,142 species, from a diverse set of taxonomic categories. The list of kingdoms represented as well as the number of subcategories and number of images available are shown in Table 6.

PLOS Computational Biology

**Table 6**. Overview of the classes distribution and amount of data in the INaturalist2018 dataset.

| Kingdom | Category Count | Train Images | Val Images |
|---|---|---|---|
| **Plantae** | 2,917 | 118,800 | 8,751 |
| **Insecta** | 2,031 | 87,192 | 6,093 |
| **Aves** | 1,258 | 143,950 | 3,774 |
| **Actinopterygii** | 369 | 7,835 | 1,107 |
| **Fungi** | 321 | 6,864 | 963 |
| **Reptilia** | 284 | 22,754 | 852 |
| **Mollusca** | 262 | 8,007 | 786 |
| **Mammalia** | 234 | 20,104 | 702 |
| **Animalia** | 178 | 5,966 | 534 |
| **Amphibia** | 144 | 11,156 | 432 |
| **Arachnida** | 114 | 4,037 | 342 |
| **Chromista** | 25 | 621 | 75 |
| **Protozoa** | 4 | 211 | 12 |
| **Bacteria** | 1 | 16 | 3 |
| **Total** | 8,142 | 437,513 | 24,426 |

The classifier model [35] is a ResNet50-based Bilateral-Branch Network (BBN) architecture, pre-trained on Imagenet data and fine-tuned on the training subset of the INaturalist2018 dataset.

## 600k Dataset definition, annotation and analysis

To exploit the full potential of the models we presented earlier, we collected from the Consortium of Northeastern Herbaria Portal [72] a massive dataset in order to analyse the flowering patterns of a large number of species and specimens from accross northeastern North America. We first set a criterion for admissible species, such as possessing stereotypical flowers, excluding families in groups like graminoids and ferns for example. The flowers also had to be large enough to be identified directly from the digitized sample, thus excluding families that usually have microscopic flowers.

We identified over 600,000 specimens spread over 19,000 species matching these constraints. After performing taxonomic reconciliation against the USDA Plant database for accepted species names, we obtained around 4000 final species. The reconciliation was performed using the R package U. Taxonstand [73].

**Performances analysis against manually annotated ground-truth.** In order to validate the approach against a human annotated ground-truth, a subset of the dataset was annotated manually. The subset contained 15,000 specimens representing 20 species.

This data was used to validate the performance of the regression process: for each species, we performed a linear regression to determine the shift of its flowering season, both from the human-annotated data, and from the model at several confidence threshold levels. One such example of trend estimation can be seen in Fig 17. By investigating the factors impacting the error slope, we can see on Fig 18 that the threshold choice does not seem to have a noticeable impact, but the amount of samples used to perform the linear regression seems critical. The error increases exponentially as the number of samples decreases. These two observations can be tied to the nature of the estimation task that we perform with our data: a linear regression is sensitive to noise, and our data is very sparse with regard to flowering time information. This type of task benefits more from having a large quantity of data than having a sparser set of higher quality data to obtain a reliable estimate. This will guide our choices for exploiting our data for this use-case:

- always use the estimate from the 50% minimum threshold to retain as many data points as possible
- filter out any species that has less than 75 samples (based on Fig 18)

Our objective for this study being to estimate the flowering time shift between pre-industrial and post-industrial times, we can only do that reliably by having a minimum spread of data around the industrial tipping point (which we take as 1950).

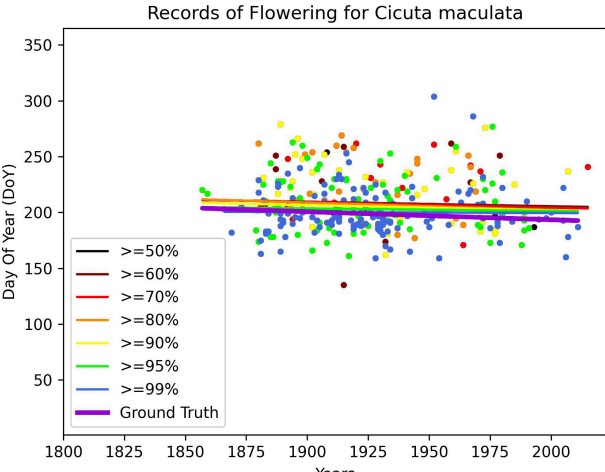

**Fig 17**. **Model annotation estimated trends for each threshold (Black, Brown, Red, Orange, Yellow, Green, Blue), human annotations estimated trend (Purple).**

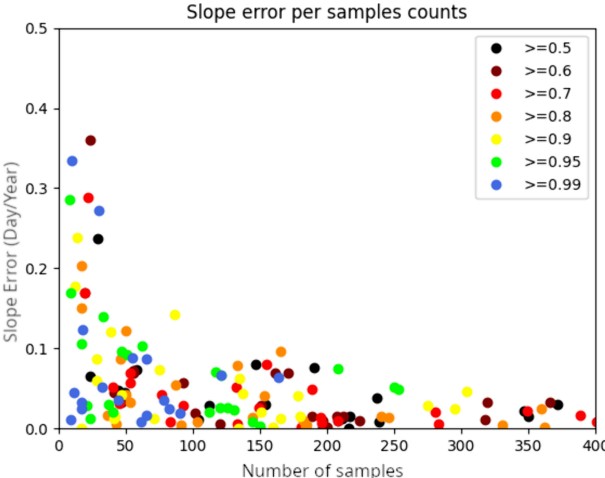

**Fig 18**. **Amount of slope error (between the automatically annotated and the human annotated ground-truth) against the number of available samples to perform the regression.** Each dot corresponds to a given species for a given threshold.

To enforce that condition, we introduce a final filter to remove any species that doesn't have at least 37 (half of the total minimum of samples) samples before and after 1950.

**Subset analyses.** For each characteristic we assigned species to categories, and within each category, we examined the change in flowering time in days per year at the 50% confidence level. Shifts were calculated for each species within a category and then averaged across all species. We then used Welsh's T-test to evaluate for each characteristic if there were statistically significant differences in flowering time shift among categories.

**Growth form subset.** Growth forms were assigned using the USDA PLANTS Database [74]. We categorized a subset of species into one of three lifeform categories: forb/herb, tree/shrub/subshrub, or vine. A forb/herb is defined as a vascular plant without significant woody tissue above or at the ground. A tree/shrub/subshrub is a vascular plant with a perennial wood stem(s). Vines are defined as a twining/climbing vascular plants with relatively long stems, and can be woody or herbaceous. For further information, consult the USDA Plants Database help document [75].

**Nativity subset.** We categorized species as native or introduced to New England states using the USDA PLANTS Database classifications [74]. In the PLANTS database, "Native" refers to plants naturally occurring in the floristic area in the late 15th Century, the beginning of persistent European colonization in North America. "Introduced" plants are defined as those that arrived, with human assistance, in the floristic area at that time or later from outside of the floristic area.

**Wetland status subset.** We categorized a subset of species by their National Wetland Plant List (NWPL) Wetland Indicator Status within the Northcentral and Northeast Region [76] using categorizations provided by the USDA PLANTS Database [74]. There are five indicator statuses that indicate a plant species' preference for occurrence in a wetland or upland (Table 7). For further information, including details about the methodology underlying the classifications, consult the official NWPL website [77].

**Seasonal timing of flowering.** Based on flowering times discerned from specimen data we categorized species as early season if the mean DOY was 180 (ca. June 29) or less and late season if mean DOY was greater.

**Flowering duration.** We categorized species as having a narrow or broad flowering duration. Species with a narrow duration were defined as those that were found to have an average flowering duration of 28 days or less and those with a broad range had an average flowering duration greater than that.

**Phylogenetic analyses.**

**Phylogeny and data reconciliation.** We utilized the 353,185 taxa of seed plants from [79] to explore phylogenetic patterns in our dataset. Briefly, we reduced our 680-taxon dataset to 648 taxa by removing subspecies and entries that were resolved to genus only. For species with more than one subspecies included, we retained the subspecies with the most collection points. We then matched our 648 taxon names to the 353,185 taxon names in the phylogeny and recovered an overlap of 570 taxa. We then pruned the large tree to a 570-tip tree and proceeded with phylogenetic analyses for this reduced dataset. We did not include wetland status in these analyses as many taxa were not included for this character. We also slightly changed the "growth form" character to woody vs. herbaceous as we needed binary character states for some analyses.

**Phylogenetic signal.** We first explored whether significant flowering shifts or any of our biological variables (growth form, native status, flowering seasonality, flowering duration) exhibited phylogenetic signal. Because our characters are structured as binary traits, we inferred the strength of phylogenetic signal by estimating values of lambda [80], using the "fitDiscrete" function in the Geiger R package [81], where lambda values of 0 indicate no signal and values of 1 indicate high signal.

**Trait correlations.** We identified 4 biological variables that we thought might influence whether or not a species experienced a significant phenological shift. To examine their potential association within a phylogenetic framework, we utilized

**Table 7**. **National Wetland Plant List Wetland Indicator Status categories and definitions.** See [78] for more in-depth definitions and other information.

| Indicator Status | Abbreviation | Definition |
| --- | --- | --- |
| Obligate | OBL | Almost always occurs in wetlands |
| Facultative Wetland | FACW | Usually occurs in wetlands, but may occur in non-wetlands |
| Facultative | FAC | Occurs in wetlands and non-wetlands |
| Facultative Upland | FACU | Usually occurs in non-wetlands, but may occur in wetlands |
| Upland | UPL | Almost never occurs in wetlands |

the method of [82], which compares two models of trait evolution: one in which transitions between two character states are independent in both characters, vs a model where transition rates in one character are dependent on the character state of a second character. In our analyses, the potentially "dependent" character was phenological shift, and the independent character was the biological variable. We tested phenological shifts grouped together as well as early shifts and late shifts separately. For each test, we compared maximum-likelihood scores of each model with a likelihood-ratio test to assess which model better fit the data. All analyses were performed in the Geiger R package [81].

## Author contributions

**Conceptualization:** Quentin Bateux, Jonathan Koss, Patrick W. Sweeney, Nelson Rios, Aaron M. Dollar.

**Data curation:** Quentin Bateux, Patrick W. Sweeney.

**Formal analysis:** Quentin Bateux, Erika Edwards.

**Investigation:** Quentin Bateux, Erika Edwards.

**Methodology:** Quentin Bateux, Patrick W. Sweeney, Erika Edwards, Nelson Rios, Aaron M. Dollar.

**Project administration:** Quentin Bateux.

**Software:** Quentin Bateux, Jonathan Koss.

**Supervision:** Patrick W. Sweeney, Erika Edwards, Nelson Rios, Aaron M. Dollar.

**Validation:** Aaron M. Dollar.

**Writing – original draft:** Quentin Bateux, Patrick W. Sweeney.

**Writing – review & editing:** Quentin Bateux, Patrick W. Sweeney, Erika Edwards, Aaron M. Dollar.

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
