## [Decision Letter · Decision Letter 0]

13 Apr 2025

PCOMPBIOL-D-25-00519

Improving the accuracy of automated labeling of specimen images datasets via a confidence-based process

PLOS Computational Biology

Dear Dr. Bateux,

Thank you for submitting your manuscript to PLOS Computational Biology. After careful consideration, we feel that it has merit but does not fully meet PLOS Computational Biology's publication criteria as it currently stands. Therefore, we invite you to submit a revised version of the manuscript that addresses the points raised during the review process.

Please submit your revised manuscript within 60 days Jun 13 2025 11:59PM. If you will need more time than this to complete your revisions, please reply to this message or contact the journal office at ploscompbiol@plos.org. Please include the following items when submitting your revised manuscript:

We look forward to receiving your revised manuscript.

Kind regards,

Jinshan Xu

Academic Editor

PLOS Computational Biology

Zhaolei Zhang

Section Editor

PLOS Computational Biology

**Additional Editor Comments :**

The results used to validate the effectiveness of the proposed mehtod were calculated on a dataset without the discarded sample. This makes the results questionable. The authors are encouraged to the effectiveness of the proposed method in general sence.

**Journal Requirements:**

2) We note that the following section: Discussion is mentioned twice in the manuscript  Please check the manuscript sections. An outline of the required sections can be consulted in our submission guidelines here:

4) Please upload a copy of Figures Fig. VII-A(a), and Fig. VII-A(b) which you refer to in your text on page 11. Or, if the figures are no longer to be included as part of the submission please remove all reference to them within the text.

5) Please ensure that all Figure files have corresponding citations and legends within the manuscript. Currently, Figures 12, 13, and 15 in your submission file inventory do not have in-text citations. Please include the in-text citations of the figures.

Potential Copyright Issues:

i) Please confirm (a) that you are the photographer of 1, 2, and 8, or (b) provide written permission from the photographer to publish the photo(s) under our CC BY 4.0 license.

ii) Figure 8. Please confirm whether you drew the images / clip-art within the figure panels by hand. If you did not draw the images, please provide (a) a link to the source of the images or icons and their license / terms of use; or (b) written permission from the copyright holder to publish the images or icons under our CC BY 4.0 license. Alternatively, you may replace the images with open source alternatives. See these open source resources you may use to replace images / clip-art:

7) Please number your tables with Arabic numerals and change the citations accordingly.

8) Please provide a completed 'Competing Interests' statement, including any COIs declared by your co-authors. If you have no competing interests to declare, please state "The authors have declared that no competing interests exist". Otherwise please declare all competing interests beginning with the statement "I have read the journal's policy and the authors of this manuscript have the following competing interests:"

**Reviewers' comments:**

Reviewer's Responses to Questions

Reviewer #1: This paper introduces a confidence threshold framework for deep learning models, which achieves arbitrarily high annotation accuracy by sacrificing a portion of available samples. The authors conducted extensive experiments across multiple datasets, including a new dataset containing 600,000 specimens, and performed detailed analyses. The research aligns with the scope of “PLOS Computational Biology”, but the authors should consider addressing the following issues:

1. The paper proposes a deep learning model confidence threshold framework that achieves arbitrarily high accuracy by sacrificing some samples. The accuracy reported here is calculated based on the discarded samples, which, in contrast to traditional accuracy computed across all classified samples, is essentially closer to precision. The authors should clarify the fundamental differences and advantages between the proposed approach and commonly used metrics, such as the ROC curve, which typically sacrifices recall to enhance precision (or vice versa).

2. Furthermore, the confidence output and prediction accuracy of deep neural networks can be effectively calibrated during training. Related techniques are discussed in the work of Moloud Abdar et al. in “A Review of Uncertainty Quantification in Deep Learning: Techniques, Applications, and Challenges”, which highlights numerous methods for achieving well-calibrated predictions. The authors are encouraged to consider incorporating calibration methods into the existing framework to enable more direct and accurate confidence threshold selection.

3. The experiments are conducted on fixed datasets without performing cross-validation. Are the reported metrics based on a single training run? The authors should provide further clarification on how they mitigate the potential for result variability.

Additional Comments:

1. Given that the results and discussion are already included in Section V, it is recommended that Section VI be renamed from "DISCUSSION" to "CONCLUSION".

2. Fig. 15 requires revision as the data points are heavily overlapping. Additionally, the legend in subplot (a) is unclear.

3. The header position in Table VI is inconsistent with other tables.

4. Figure 16 also suffers from heavy overlap between points and lines.

5. The "Definition" column in Table VII appears disorganized.

Reviewer #2: This manuscript reports the improved accuracy of automated labeling, mainly in the context of plant specimens, by the application of confidence threshold value. The confidence threshold value rejects the image with low machine learning (ML) labeling confidence. This increases the accuracy of the model at the cost of decreasing the coverage.

As mentioned in the manuscript, manual labeling of the samples is time-consuming, but the automated labeling should be of high accuracy for usage in scientific research. Increasing accuracy without making a new ML model should allow non-ML experts in the field to improve their label preparation efficiency. In the proposed approach, there is always a trade-off between accuracy and coverage, but this aspect is duly mentioned in the manuscript. Also, the authors showed through multiple examples that they could appropriately balance accuracy and coverage for scientific inference.

Therefore, the reviewer believes that this work has the potential to benefit the community, especially the botanists, by reducing the manual labor and speeding up the research workflow.

However, the reviewer realized several points that should be addressed for the manuscript to be accepted for publication as listed below.

Major Points

1. Examples of rejected and accepted images

The authors illustrate in Fig. 15 the distribution of low-confidence images that are likely to be rejected by confidence threshold approach. However, this distribution is derived from ML layers, and they do not show actual images. The readers will be interested in whether the confidence level goes along well with human perception of ambiguous or clear flowering judgments. Therefore, the reviewer recommends adding example images for each confidence level (corresponding to green, yellow,…,black) and flowering/non-flowering.

2. Visualization for multi-class results

The reviewer recommends adding visualization similar to Fig. 15 for the multi-class results presented in Section V-C because the characteristics for the rejected images will be one of the main interests when using this approach.

Minor Points

3. Demo code

The reviewer recommends providing a demo code for this approach so that readers can interact with the confidence thresholding effect.

4. Miscellaneous

On page 8, Table III, the reviewer believes the 2nd column in Nativity row should be native vs non-natives. Please check it.

The reviewer spotted a mislabeling of a figure. On page 11, Fig. VII should be Fig. 15. Please fix the mislabeling.

Reviewer #3: This paper proposes a practical and reasonable method for improving the reliability of automatic labeling of specimen images by considering the prediction confidence of DNN models. The idea is easy to implement, making it particularly appealing for ecological research applications. Additionally, the research leads to novel ecological findings, such as the observation that species with a higher affinity for wetter habitats shift their phenology relatively less than those in dryer habitats.

Apart from these strengths, I have several concerns:

1) One major issue is that the method relies heavily on confidence scores produced by softmax layers. However, it is well-known that DNNs are often concerned as overconfident models. Thus, model calibration should be considered and discussed to enhance the reliability of the confidence-based method. Typical techniques include temperature scaling, isotonic regression, or calibration error metrics. I suggest the authors to consider improving the method with easy-to-implement calibration methods.

2) Selecting only high-confidence examples may introduce sampling bias, as high-confidence samples may represent only those examples with simple visual patterns. The risk of sampling bias should be quantified or at least discussed, and potential mitigation strategies could be proposed.

3) The evaluation on the 600K dataset is not convincing enough since the subset used for evaluation covers only a small portion of species. If covering all the species is too expensive, you can at least consider a more reliable strategy for sampling a subset of representative examples (e.g., Core-Set) over the 600K dataset.

Overall, I think this work makes a valuable contribution, but has a few concerns regarding reliability of the proposed method and dataset should be addressed before acceptance.

**Have the authors made all data and (if applicable) computational code underlying the findings in their manuscript fully available?**

Reviewer #1: **No: **The authors have provided the data but have not made the code publicly available.

Reviewer #2: None

Reviewer #3: Yes

PLOS authors have the option to publish the peer review history of their article (what does this mean?). If published, this will include your full peer review and any attached files.

Reviewer #1: **Yes: **Tao Chen

Reviewer #2: No

Reviewer #3: No

**Figure resubmission:**
---

## [Decision Letter · Decision Letter 1]

24 Aug 2025

PCOMPBIOL-D-25-00519R1

Improving the accuracy of automated labeling of specimen images datasets via a confidence-based process

PLOS Computational Biology

Dear Dr. Bateux,

Thank you for submitting your manuscript to PLOS Computational Biology. After careful consideration, we feel that it has merit but does not fully meet PLOS Computational Biology's publication criteria as it currently stands. Therefore, we invite you to submit a revised version of the manuscript that addresses the points raised during the review process.

Please submit your revised manuscript within 30 days Oct 24 2025 11:59PM. If you will need more time than this to complete your revisions, please reply to this message or contact the journal office at ploscompbiol@plos.org. Please include the following items when submitting your revised manuscript:

We look forward to receiving your revised manuscript.

Kind regards,

Jinshan Xu

Academic Editor

PLOS Computational Biology

Zhaolei Zhang

Section Editor

PLOS Computational Biology

**Additional Editor Comments:**

Please revise the manuscript according to reviewers' comments.

**Journal Requirements:**

1) Kindly revise your competing statement to align with the journal's style guidelines: 'The authors declare that there are no competing interests.'

**Reviewers' comments:**

Reviewer's Responses to Questions

**Comments to the Authors:**

Reviewer #1: An explanation clarifying the distinction between the accuracy/recall trade-off and the accuracy/rejection trade-off has been added, together with a discussion on model calibration and related research. Appropriate revisions have also been made to address issues related to figures and formatting. The reviewer considers the manuscript suitable for publication.

Reviewer #2: My previous comment 1 is appropriately addressed. As for comment 2, I understand the authors' point, and in that case, current visualization would be enough.

As for comment 3, even if authors do not have time to make a full tutorial, I believe they should at least make their code used in the study open with minimal description for the reproducibility of the study.

Reviewer #3: Thank you for addressing the concerns. Please add a small paragraph to discuss the potential risk of data bias by selecting high-confidence examples.

**Have the authors made all data and (if applicable) computational code underlying the findings in their manuscript fully available?**

Reviewer #1: **No: **The authors have provided the data; however, the code is not included.

Reviewer #2: **No: **Data is available but code is not.

Reviewer #3: **No: **Code not provided

PLOS authors have the option to publish the peer review history of their article (what does this mean?). If published, this will include your full peer review and any attached files.

Reviewer #1: **Yes: **Tao Chen

Reviewer #2: No

Reviewer #3: No

**Figure resubmission:**

After uploading your figures to PLOS’s NAAS tool - https://ngplosjournals.pagemajik.ai/artanalysis NAAS will process the files provided and display the results in the "Uploaded Files" section of the page as the processing is complete. If the uploaded figures meet our requirements (or NAAS is able to fix the files to meet our requirements), the figure will be marked as "fixed" above. If NAAS is unable to fix the files, a red "failed" label will appear above. When NAAS has confirmed that the figure files meet our requirements, please download the file via the download option, and include these NAAS processed figure files when submitting your revised manuscript.
---

## [Decision Letter · Decision Letter 2]

22 Oct 2025

Dear Mr Bateux,

We are pleased to inform you that your manuscript 'Improving the accuracy of automated labeling of specimen images datasets via a confidence-based process' has been provisionally accepted for publication in PLOS Computational Biology.

Best regards,

Jinshan Xu

Academic Editor

PLOS Computational Biology

Zhaolei Zhang

Section Editor

PLOS Computational Biology

Reviewer's Responses to Questions

**Comments to the Authors:**

Reviewer #1: The revised manuscript now includes concise companion code (one notebook and one analysis script), which enables rapid reproduction of the exemplar figures. This addition substantially improves transparency and reproducibility.

That said, the repository would benefit from several non-blocking enhancements: (i) an explicit environment specification (e.g., requirements.txt/environment.yml or a Docker image); and (ii) an open-source license and versioned, tagged releases.

With respect to the scientific content of the article itself, I have no additional concerns. I recommend acceptance, with the above repository improvements addressed during production.

Reviewer #2: The authors addressed all the points raised.

Reviewer #3: My concern has been well addressed and I recommend acceptance of the paper.

**Have the authors made all data and (if applicable) computational code underlying the findings in their manuscript fully available?**

Reviewer #1: Yes

Reviewer #2: Yes

Reviewer #3: **No: **Code is provided but it's based on the CIFAR-10 data. Please consider to release the code for directly reproducing the results if the actual dataset can be made public.

PLOS authors have the option to publish the peer review history of their article (what does this mean?). If published, this will include your full peer review and any attached files.

Reviewer #1: **Yes: **Tao Chen

Reviewer #2: No

Reviewer #3: No

---

## [Editor Report · Acceptance letter]

PCOMPBIOL-D-25-00519R2

Improving the accuracy of automated labeling of specimen images datasets via a confidence-based process

Dear Dr Bateux,

I am pleased to inform you that your manuscript has been formally accepted for publication in PLOS Computational Biology. Your manuscript is now with our production department and you will be notified of the publication date in due course.

With kind regards,

Judit Kozma
